# Efficacy of Early Mobilization in Stroke Patients in Relation to Quality of Life and Level of Dependency: A Systematic Review

**DOI:** 10.3390/healthcare14010078

**Published:** 2025-12-29

**Authors:** Malena Maffassanti-Reyes, Marta González-Sierra, Alberto Javier-Ormazábal

**Affiliations:** 1Division of Physiotherapy, Hospital Universitario de Canarias, Carretera Ofra S/N, 38320 San Cristóbal de La Laguna, Santa Cruz de Tenerife, Spain; mmafrey@gobiernodecanarias.org; 2Division of Internal Medicine, Hospital Universitario de Canarias, Carretera Ofra S/N, 38320 San Cristóbal de La Laguna, Santa Cruz de Tenerife, Spain; marta.gonzalez@universidadeuropea.es; 3Physiotherapy Department, Universidad Europea de Canarias, C. Inocencio García, 1, 38300 La Orotava, Santa Cruz de Tenerife, Spain; 4Department of Physical Medicine and Pharmacology, Universidad de La Laguna, C/Sta. María Soledad, S/N, 38071 San Cristóbal de La Laguna, Santa Cruz de Tenerife, Spain

**Keywords:** stroke, early mobilization, quality of life, dependency

## Abstract

**Introduction:** Stroke is a severe neurological condition associated with high rates of mortality and disability. **Objective:** This systematic review aimed to analyze the efficacy of early mobilization (EM) on the quality of life and the dependency levels in stroke patients. Additionally, the impact on anxiety and depression, the occurrence of adverse effects, and length of hospital stay were assessed. **Methods:** A systematic search was conducted in PubMed, Web of Science, and Scopus databases. The search was restricted to randomized controlled trials published within the last 10 years that included EM as an intervention in the experimental group. The Cochrane tools were used to assess risk of bias, and the PEDro scale was applied to evaluate the methodological quality of the included studies. **Results:** Nine studies were included in this review. Findings indicated that EM performed within 24–48 h post-stroke reduces dependency levels; however, no significant improvement in quality of life was observed. Evidence regarding anxiety and depression was inconclusive, and no significant differences were reported between groups concerning adverse events or reduction in hospital stay duration. **Conclusions:** This review demonstrates that EM is beneficial for reducing dependency after stroke, but there is no evidence of a significant improvement in quality of life. Further research is needed to establish clear protocols and appropriate intervention doses.

## 1. Introduction

A stroke is defined as a sudden disruption of cerebral circulation that causes a transient or permanent alteration of brain function. The underlying mechanism determines diagnostic tests, acute and preventive treatment, and prognosis [1,2]. Strokes are classified as ischemic or hemorrhagic. Ischemic stroke, the most common type (80–85%), results from an interruption of blood supply due to vessel occlusion. It can be global or focal and is further categorized into transient ischemic attack (TIA) or cerebral infarction [3,4,5]. TIAs resolve within 24 h, whereas infarctions persist beyond 24 h and involve tissue necrosis [5,6]. Etiologically, 50% are due to large vessel atherothrombosis, 25% to cardioembolic, 20% to lacunar infarcts, and 5% to other causes [5,6]. Hemorrhagic stroke, accounting for 15–20%, involves vessel rupture and bleeding into brain tissue or subarachnoid space [5,6].

Stroke incidence has risen significantly, being the leading cause of death among cardiovascular diseases in Spain and the primary cause of disability in Western countries. It predominantly affects individuals aged 60–65, though cases in younger populations are increasing [3,4]. Stroke is a neurological emergency requiring rapid treatment. Symptoms vary by lesion site and severity, including facial asymmetry, hemiparesis, speech difficulties, imbalance, and severe headache. Consequences include motor deficits, sensory loss, gait impairment, language disorders, dysphagia, visual field deficits, behavioral changes, anxiety, and depression. Approximately 44% of survivors develop significant functional dysfunction [7,8,9,10].

The “stroke code” protocol organizes early diagnosis and individualized treatment, reducing morbidity and mortality [8]. Stroke units provide specialized care, focusing on rapid diagnosis, stabilization, and prevention of complications through strict blood pressure control, glycemia management, and temperature regulation. Early rehabilitation is recommended [11,12]. Neuroimaging advances enable precise evaluation: non-contrast CT is first line for hemorrhagic stroke, while diffusion MRI offers higher sensitivity for ischemic stroke. Perfusion imaging identifies salvageable tissue [13,14].

Treatment is multidisciplinary, involving neurologists, physiatrists, neuropsychologists, rehabilitation therapists such as physiotherapists, occupational therapists and speech therapists [2,11,15]. For ischemic stroke, reperfusion via thrombolysis or mechanical thrombectomy within 3–5 h are standard; beyond this window, pharmacological management and stenting may be used [11,15]. Hemorrhagic stroke treatment aims to control bleeding and intracranial pressure through antihypertensives, hemostatics, diuretics, hypertonic solutions, and sometimes surgery [15,16].

Rehabilitation, essential from early to chronic phases, is underpinned by neuroplasticity and requires individualized physiotherapy goals. In the hyperacute phase (24–72 h), stabilization is prioritized; later phases focus on functional recovery and independence. Techniques include therapeutic exercise, task-oriented motor relearning, constraint-induced therapy, mirror therapy, proprioceptive neuromuscular facilitation, and advanced technologies such as virtual reality, robotics, and functional electrical stimulation [17,18,19,20,21]. Recent evidence strongly supports early mobilization in patients with various conditions, including those in intensive care units and stroke units [22]. Early mobilization refers to the prompt initiation of verticalization, sitting, standing, and walking, typically within 24–48 h after stroke onset. This approach promotes neuroplasticity by activating collateral cortical areas and forming new neuronal connections [23,24].

Although early mobilization is superior to conventional physiotherapy techniques, its implementation must consider stroke severity, hemodynamic stability, and comorbidities. Current literature suggests greater benefits in patients with mild strokes compared to severe cases [25,26]. Numerous studies have shown that early mobilization accelerates motor recovery, reduces secondary complications such as deep vein thrombosis and pressure ulcers, improves cerebral blood flow, enhances respiratory function, and activates unaffected brain regions. Additionally, it appears to lower healthcare costs [25,27,28].

Given the high incidence and prevalence of stroke, and its increasing occurrence in younger populations, optimizing rehabilitation protocols is essential. Physiotherapists play a critical role in this process. Early mobilization has gained prominence due to robust scientific evidence supporting its impact on functional recovery. Early physiotherapy fosters neuroplasticity and reduces complications associated with prolonged hospitalization, while also decreasing short- and long-term healthcare costs by promoting faster recovery and reducing functional impairment [29,30,31,32,33].

Quality of life (QoL) in stroke patients is closely linked to their ability to perform daily activities and their emotional and social well-being. Early mobilization may significantly improve QoL and reduce dependency, lowering the risk of immobility-related complications such as pressure ulcers, deep vein thrombosis, and pneumonia, while enhancing psychological well-being.

This study aims to evaluate the efficacy of early mobilization in stroke patients, focusing on its relationship with QoL and functional recovery measured by dependency level. Positive results would not only confirm clinical benefits but also demonstrate its potential to reduce the healthcare and socioeconomic burden associated with stroke in the long term.

## 2. Materials and Methods

This systematic review was conducted in accordance with Cochrane Collaboration recommendations and reported following the PRISMA 2020 statement to synthesize current evidence on early mobilization in stroke patients regarding quality of life and dependency levels. PRISMA includes a 27-item checklist and a four-phase flow diagram [34]. The review was registered with the Open Science Framework (OSF) with the registration code https://osf.io/kvug4 (accessed on 1 December 2025).

### 2.1. Eligibility Criteria

Selection criteria were based on methodological and clinical aspects structured using the PICOs strategy: Population, Intervention, Comparison, Outcomes, and Study type. The research question was: In patients with acute stroke, does early mobilization (initiated within 24–72 h post-stroke) compared to standard care or later mobilization improve dependency level, quality of life, mental health, and reduce complications and hospital stay.

### 2.2. Population

Included studies involved patients diagnosed with acute stroke, without restrictions on age, sex, stroke type, or severity. Studies with patients having other severe neurological conditions or comorbidities interfering with recovery were excluded.

### 2.3. Intervention and Comparison

Studies compared early mobilization initiated within 24–72 h post-stroke in the experimental group versus standard care or later mobilization in the control group. Early mobilization was defined as active sitting, standing, and walking whenever possible. Studies combining early mobilization with other techniques were excluded to avoid bias.

### 2.4. Variables

Primary outcomes were quality of life and dependency level. Quality of life was assessed using validated tools such as the Stroke Impact Scale and Assessment of Quality of Life (AQoL), which measure physical, emotional, and social well-being [35,36,37,38]. Dependency was evaluated using the Modified Rankin Scale and Barthel Index, both widely validated instruments [39,40,41].

### 2.5. Methodology Overview

The Barthel Index is a simple, reliable, and valid tool widely used to assess a person’s ability to perform daily living activities, scoring from 0 (total dependence) to 100 (complete independence) [42,43]. Secondary variables considered in this review included hospital stay duration, adverse events (such as stroke progression, recurrent stroke, pulmonary embolism, deep vein complications, urinary tract infection, pressure ulcers, pneumonia), and levels of anxiety and depression.

### 2.6. Study Types

Only randomized controlled trials published in the last 10 years were included. Quasi-experimental studies, case series, and reviews were excluded. Articles scoring below 6 on the PEDro scale were discarded to ensure good methodological quality. Based on these conditions, the PICOS (patient population, interventions or exposure, comparator group, outcome or endpoint, and study design) search tool was employed to generate keywords (Table 1) for the systematic search of electronic publication databases:

### 2.7. Information Sources and Search Strategy

A systematic search was conducted on April and October 2025, in PubMed, Scopus, and Web of Science to identify recent studies analyzing the effects of early mobilization in acute stroke patients, specifically regarding quality of life and dependency. Articles published between 2015 and 2025 were considered. Search strategies combined MeSH terms and keywords related to stroke, early mobilization, and study type, using Boolean operators (“AND”, “OR”) adapted to each database. The following table (Table 2) presents the consulted sources and the exact search equations used.

### 2.8. Data Extraction

Following Cochrane recommendations, potentially eligible studies were screened by title and abstract, then reviewed in full text. Data extracted included author and year, participant characteristics (age, sex, stroke type), sample size, interventions and dosage for experimental and control groups, variables assessed, timing of evaluations, and results.

### 2.9. Methodological Quality and Risk of Bias

The PEDro scale was used to assess methodological quality, scoring 10 items related to internal validity and statistical analysis. According to the PEDro scale studies were classified as low (<4), fair (4–5), good (6–8), or excellent (9–10). All studies with a methodological quality score of less than 6 were excluded from this systematic review [44]. Risk of bias was evaluated using Cochrane’s tool across six domains: randomization, allocation concealment, blinding, incomplete outcome data and selective reporting. Each domain was rated as low, high, or unclear risk, with justification provided [45].

## 3. Results

### 3.1. Study Selection

A total of 387 articles were identified, of which 155 duplicates were removed, yielding 232 unique records. An initial screening of titles and abstracts resulted in the exclusion of 193 articles. Subsequently, the full texts of the 10 articles that met the selection criteria were reviewed [24,46,47,48,49,50,51,52,53,54]. One article was excluded for not achieving a score of 6 or higher on the PEDro scale [54]. Ultimately, 9 articles met all eligibility criteria and were included in the present review (Figure 1).

### 3.2. Characteristics of Study Populations

All included patients had experienced a stroke. The total number of patients across the included studies was 6603, with sample sizes ranging from 40 to 2104 participants. Regarding sociodemographic variables, the mean age was 64 ± 6.4 years, and all studies included both men and women.

With respect to stroke severity, 8 of the 9 articles used the NIHSS scale. In two of the included studies by Chippala et al. (2016) [53] and Wang et al. (2022) [49], most patients presented moderate NIHSS scores (8–16 points). However, in the studies by Cumming et al. (AVERT group, 2015) [47], Langhorne et al. (2017) [46], Anjos et al. (2022) [52], and Tong et al. (2019) [48], approximately 70% of participants had mild initial NIHSS scores (0–7 points). Yelnik et al. (2017) [50] included a population with a high proportion of severe strokes. The study by Rahayu et al. (2019) [51] used the ASPECTS scale instead of NIHSS to assess stroke severity, where ASPECTS < 7 indicated mild stroke severity.

Regarding stroke type, all participants in four studies had ischemic stroke [48,49,51,52]. The remaining five studies included populations composed of approximately 80–90% ischemic stroke and 10–20% hemorrhagic stroke [24,46,47,50,53].

### 3.3. Study Characteristics

Of the 9 included articles, 8 allocated participants into two groups—an experimental group and a control group—differentiating the timing of early mobilization onset and the prescribed intervention dose [24,46,47,48,49,51,52,53]. Early mobilization was defined as active mobilization of all four limbs, standing, balance exercises, and gait training whenever possible. In the following table (Table 3), the characteristics of the studies included in this review are presented.

### 3.4. Assessment of Methodological Quality

The methodological quality of the studies was evaluated using the PEDro scale, and the results are presented in Table 4. The mean score of the studies included in this systematic review was 7.22 points, ranging from 6 to 10 points, indicating that the overall methodological quality was good.

### 3.5. Risk of Bias Assessment

#### 3.5.1. Random Sequence Generation and Allocation Concealment

All included articles reported that the sample had been randomized. In one study, details on the randomization procedure used to ensure allocation concealment were not provided, although the authors stated that concealment had been performed [51]. In the study by Anjos et al. (2022) [52], baseline NIHSS scores differed between the control and experimental groups, with lower scores in the experimental group. This imbalance could have influenced the results and should be considered when interpreting the lack of superiority observed for very early mobilization. Overall, a low risk of bias was established for the random allocation of participants.

#### 3.5.2. Blinding of Participants and Personnel

Regarding participant blinding, three articles implemented single blinding, where participants were unaware of their intervention group [24,46,53]. In the AVERT group (2015) [47] study, investigators were not blinded, but participants and healthcare personnel were masked. Anjos et al. (2022) [52] reported no blinding of either participants or researchers. In the remaining four studies, due to the nature of the interventions, both patients and evaluators were aware of the allocated intervention; therefore, these articles were considered at high risk of bias [48,49,50,52]. Rahayu et al. (2019) [51] did not provide information on participant or personnel blinding.

#### 3.5.3. Blinding of Outcome Assessors

Among all studies, seven implemented rigorous assessor blinding and used appropriate statistical methods [24,46,47,48,49,50,53]. Anjos et al. (2022) [52] described an open-label study without assessor blinding, representing a high risk of bias. Rahayu et al. (2019) [51] did not explicitly report assessor blinding, which poses a potential detection bias that may affect result validity.

#### 3.5.4. Incomplete Outcome Data

Most studies conducted intention-to-treat analyses and reported no loss to follow-up, indicating a low risk of bias. Only two articles reported participant attrition. Anjos et al. (2022) [52] reported a high attrition rate at 90 days—approximately 48% of the sample—due to the COVID-19 pandemic. Tong et al. (2019) [48] reported a dropout rate of nearly 15%, which represents a high risk of bias.

#### 3.5.5. Outcome Measurement

Although the outcome measurement methods were appropriate in most studies, two articles presented potential measurement bias. Anjos et al. (2022) [52] described an open-label design without assessor blinding, which could interfere with the assessment and interpretation of outcomes. Rahayu et al. (2019) [51] did not specify whether assessors were blinded, suggesting possible measurement bias that could affect outcome evaluation.

### 3.6. Other Sources of Bias

Langhorne et al. (2017) [46] reported modifications in the timing of intervention initiation in the control group over the course of the trial, making it more similar to the experimental group. This could have influenced the results and represents a potential source of bias.

Figure 2 and Figure 3 present the overall risk of bias of the included studies. Figure 4 and Figure 5 show the risk of bias for each study individually, categorized according to the type of statistical analysis used.

### 3.7. Synthesis of Results by Variables

The results of this review are presented independently for each of the variables of interest described in the methodology.

#### 3.7.1. Quality of Life

Based on the findings and the methodological quality of the included studies, we can conclude that there is scientific evidence that early mobilization does not modify the quality of life of post-stroke patients. Of all the articles included in this review, only three assessed the quality of life (QoL) variable, and all reported no statistically significant differences between early mobilization after stroke and QoL outcomes. It is noteworthy that QoL was assessed after 3 months in the study by Yelnik et al. (2017) [50] and at 12 months in the remaining two studies [24,46]. Moreover, different assessment tools were used, including the Stroke Impact Scale, the Assessment of Quality of Life, and the Assessment of Quality of Life-4 Domains [24,46,50].

#### 3.7.2. Level of Dependence

Regarding the level of dependence, all studies except Cumming et al. (2019) [24] evaluated this variable in relation to early mobilization (EM). Two outcome measures were used: the Modified Rankin Scale in six studies [46,47,48,49,50,52] and the Barthel Index in two studies [51,53]. The findings across studies were heterogeneous and dependent on the timing of reassessment and the timing of EM initiation following stroke (See Table 3).

Rahayu et al. (2019) [51] reported a positive correlation within 7 days between EM and a lower level of dependence, but it had a small sample size (n = 40). At 3 months post-treatment, three studies found no significant differences between groups [46,50,52]. However, in other studies in which retesting also occurred at 3 months, significant improvements were observed in the level of dependence associated with EM [24,48,49].

The only study that conducted a 12-month follow-up for this variable was Langhorne et al. (2017) [46], who reported a significant difference between the experimental and control groups in favor of early mobilization in post-stroke patients but it does not report the specific effect size. In contrast, the AVERT group (2015) [47] found a statistically significant difference favoring the control group rather than the experimental group. It should be noted that in this case the mean time to intervention onset in the experimental group was approximately 18 h post-stroke, compared with 22 h in the control group.

#### 3.7.3. Levels of Anxiety and Depression

Among the articles included in this review, there is no evidence that early mobilization reduces anxiety or depression levels. Cumming et al. (2019) [24] examined the relationship between quality of life and levels of anxiety and depression, establishing a direct association between these variables. Conversely, although Langhorne et al. (2017) [46] listed anxiety and depression among the variables to be assessed, the results section did not include a direct measurement; instead, an indirect correlation was made between the patient’s general condition and these variables.

#### 3.7.4. Adverse Effects of Early Mobilization

This variable was reported in four studies, all of which indicated no significant differences between groups, although adverse events were present in both. The most prevalent adverse events were seizures, falls, stroke progression, and pneumonia.

#### 3.7.5. Length of Hospital Stay

Anjos et al. (2022) [52] and Wang et al. (2022) [49] reported no significant differences between groups regarding early mobilization and length of hospital stay. In contrast, Chippala et al. (2016) [53] demonstrated a significant difference in favor of the experimental group, associating early mobilization with a reduction in hospital stay duration.

## 4. Discussion

The findings of this review indicate that early mobilization (EM) has a positive effect on functional dependence, although no significant improvement is observed in quality of life. Overall, the evidence suggests that EM is associated with reduced dependence levels, particularly when mobilization begins within approximately 24 h post-stroke, where significant differences at 3 months are consistently reported [46,47,48,49,53]. Rahayu et al. (2019) [51] is the only study with a short-term retest, showing early benefits at 7 days post-stroke. Two studies reporting non-significant results Yelnik et al. (2017) [50] and Anjos et al. (2022) [52] present specific contextual factors that may explain these findings: high proportions of severe stroke cases and delayed intervention (>48 h) in the former, and extremely early initiation (<12 h) combined with high fibrinolysis rates in the latter.

Physiological mechanisms may also contribute to divergent outcomes. While physical activity promotes neuroplasticity, trophic factor upregulation, and reduced apoptosis after 24 h, animal models suggest that very early activity (<10 h) may impair cerebrovascular autoregulation and worsen ischemic injury [55,56,57,58,59]. More controlled clinical trials using biomarkers and advanced perfusion imaging are needed to clarify these mechanisms.

The reviewed studies highlight treatment dose as a relevant factor for functional recovery, whereas intensity appears less influential. High-intensity protocols, such as that used by Yelnik et al. (2017) [50], did not produce superior outcomes, whereas distributed practice with short, frequent sessions yielded better functional improvements (AVERT group, 2015 [47]; Cumming et al., (2019) [24]; Langhorne et al., (2017) [46]; Tong et al., (2019) [48]. This supports the hypothesis that session distribution, rather than intensity alone, may optimize clinical outcomes.

Despite the association between functional dependence and quality of life, current evidence does not support a direct relationship between EM and improved quality of life. QoL is a complex, multidimensional construct influenced not only by physical recovery but also by psychological and social factors. Stroke survivors often experience persistent functional limitations and elevated frustration regarding loss of premorbid status, which may hinder improvements in QoL despite early physical gains. The follow-up times used in the included studies (3–12 months) and the lack of guaranteed long-term healthy lifestyle adherence after EM may further explain the absence of significant QoL differences. Risk factors such as poor diet, hypertension, diabetes, smoking, and sedentary behavior—as well as the absence of post-EM intervention protocols—may also contribute.

No conclusive results were obtained regarding levels of anxiety and depression. Cumming et al. (2019) [24] reported a direct relationship between higher QoL and lower emotional distress, consistent with the WHO’s holistic definition of health. However, no study demonstrated a direct effect of EM on emotional outcomes.

Regarding adverse effects, the reviewed studies found no significant differences between groups, aligning with current safety evidence indicating that EM does not increase the probability of complications [60,61,62,63]. Hospital length of stay also showed minimal differences (1–2 days), likely because acute stroke care protocols are standardized and time-bound. Nonetheless, even small differences may have substantial economic implications. According to the Spanish Ministry of Health’s 2024 Stroke Strategy, average post-stroke hospital stays reach 11.8 days, with ICU stays averaging 6.8 days, amounting to an average cost of €7599 per admission—approximately €410 per day. Given stroke prevalence, reducing hospitalization days or preventing admissions through stroke prevention strategies would be highly beneficial [30,64].

In Spain, the average healthcare cost per patient is €11,060 in the first year, which is €7829 more than in the year prior to the stroke. Healthcare costs decrease in the second and third year, stabilizing at approximately €340 per month around 18 months, but remaining roughly €100 above baseline (pre-stroke) levels [65]. Therefore, early mobilization could represent a significant economic saving for the national healthcare system. The data gathered in this article provide valuable insights for healthcare management.

Given the incidence of stroke, its increasing occurrence in younger populations, and the clinical relevance of early mobilization, this topic is highly current. Moreover, the specific focus on quality of life and level of dependence is essential for reducing the negative effects of this pathology. These variables are key predictors of health, and the implementation of treatment approaches that mitigate their negative impact may reduce the demand for healthcare and social-care resources in the post-stroke population. For this reason, the findings of this review may have an important impact on evidence-based rehabilitation programs and may stimulate future lines of research.

The main limitations identified in the selected studies include small sample sizes, as observed in Yelnik et al. (2017) [50], Rahayu et al. (2019) [51], and Anjos et al. (2022) [52], and considerable variability in stroke severity across samples, which may influence the results. For example, Wang et al. (2022) [49] excluded patients with severe strokes. The type of stroke included may also represent a limitation, as most participants had ischemic stroke, although this reflects its higher prevalence. Additionally, the absence of blinding of patients and therapists in some studies constitutes a risk of bias. The generalizability of our results is limited by the patient cohort, which predominantly consisted of individuals with mild to moderate ischemic stroke. Therefore, our findings may not be directly applicable to patients with severe strokes or other stroke subtypes (e.g., hemorrhagic).

Modifications to study protocols during implementation also represent a limitation; as described by the AVERT group (2015) [47] and Langhorne et al. (2017) [46], changes were made to the predefined intervention timing in the control group. Another limitation regarding the quality-of-life variable is the small number of studies that assessed it. Furthermore, insufficient follow-up durations or significant loss to follow-up pose important limitations. Yelnik et al. (2017) [50] terminated the study early due to recruitment difficulties, Rahayu et al. (2019) [51] conducted follow-up only during the first week, and Anjos et al. (2022) [52] reported a high attrition rate due to the COVID-19 pandemic.

Finally, variability in intervention protocols—particularly regarding the timing and dosage of early mobilization—constitutes an important limitation. This lack of homogeneity hinders result comparability and complicates the development of clinical practice guidelines.

## 5. Conclusions

Current evidence indicates that early mobilization, initiated approximately 24 h after stroke and delivered through short, frequent sessions, may reduce post-stroke dependency in the subacute phase. However, the substantial heterogeneity across intervention protocols limits the establishment of definitive clinical recommendations, highlighting the need for multicenter studies with clearly defined onset timing and mobilization dosage. No consistent improvements have been observed in quality of life at medium- or long-term follow-up, and the available data are insufficient to determine its effects on anxiety and depression. Early mobilization does not appear to influence hospital length of stay nor does it increase the risk of adverse events. Future research should further explore its short-term impact on quality of life and work toward standardizing intervention protocols to support the development of evidence-based clinical practice guidelines.

## Figures and Tables

**Figure 1 healthcare-14-00078-f001:**
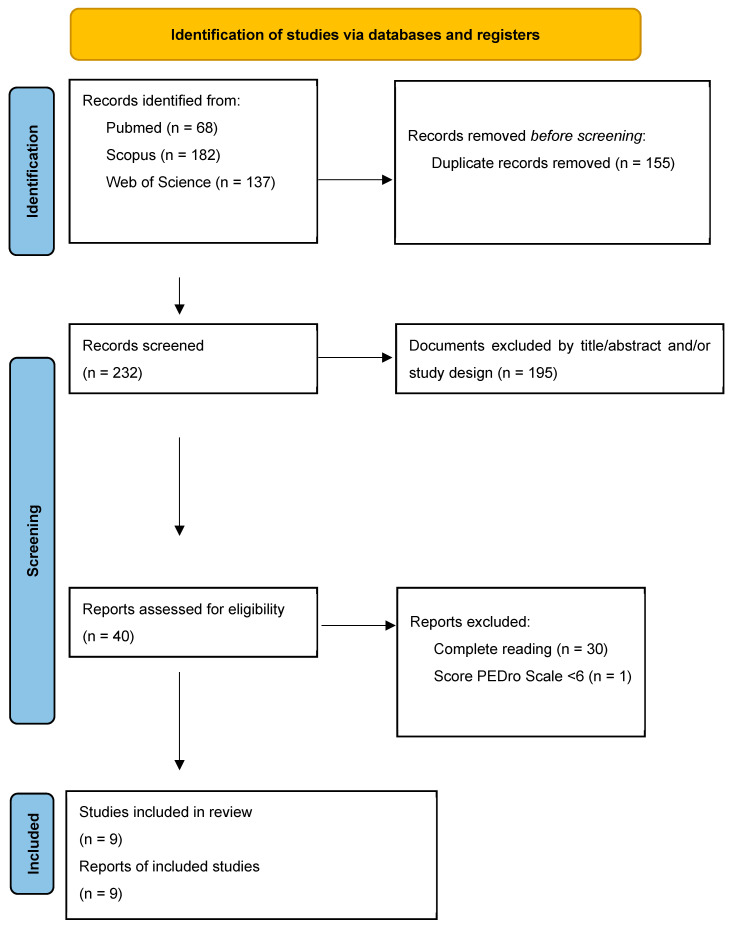
Flow Diagram.

**Figure 2 healthcare-14-00078-f002:**
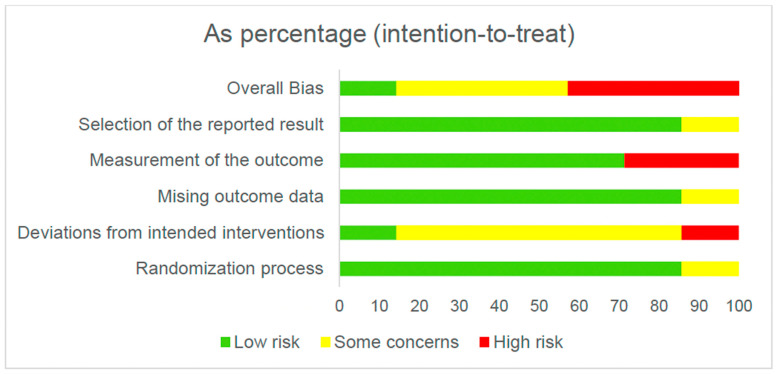
Overall risk of bias (intention-to-treat).

**Figure 3 healthcare-14-00078-f003:**
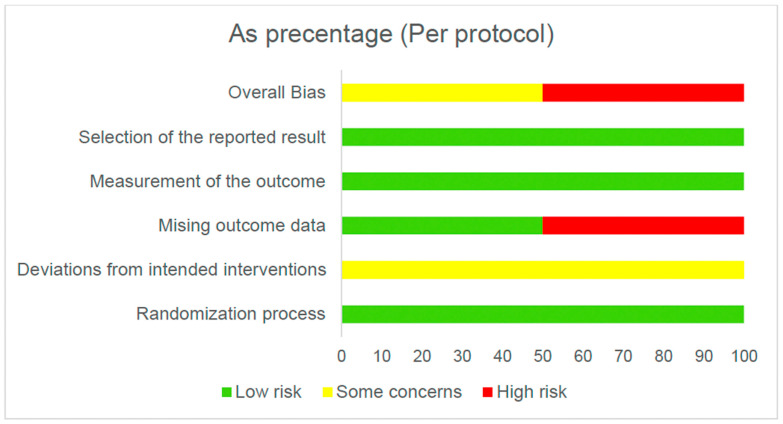
Overall risk of bias (per protocol).

**Figure 4 healthcare-14-00078-f004:**
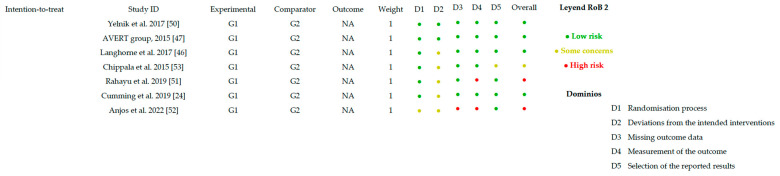
Bias risk by article (intention-to-treat) [24,46,47,50,51,52,53].

**Figure 5 healthcare-14-00078-f005:**
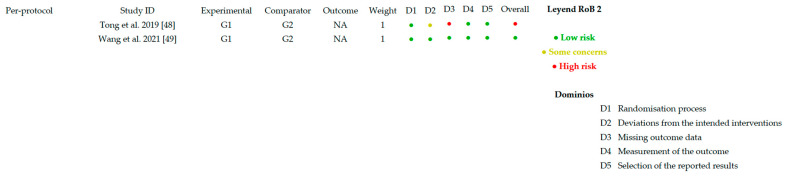
Risk of bias per article (per protocol) [48,49].

**Table 1 healthcare-14-00078-t001:** PICOS item generated keywords.

PICOS Item	Keyword
Patient Population Interventions Outcomes Study design	Stroke/cerebrovascular accident/brain infarction Early Mobilization Randomized Controlled trial

**Table 2 healthcare-14-00078-t002:** Search strategies and results by database.

Database	Exact Search Equation	Results
PubMed	(“stroke” [MeSH Terms] OR “stroke” [All Fields] OR “strokes” [All Fields] OR “cerebrovascular accident” [All Fields] OR “brain infarction” [All Fields] OR “ischemic stroke” [All Fields] OR “hemorrhagic stroke” [All Fields] OR “brain attack” [All Fields]) AND (“early mobilization” [All Fields] OR “early mobilization” [All Fields] OR “early movement” [All Fields] OR “early rehabilitation” [All Fields]) AND (“clinical trial” [Publication Type] OR “randomized controlled trial” [Publication Type] OR “randomized controlled trial” [All Fields] OR “RCT” [All Fields]) AND “humans” [MeSH Terms]	68
Scopus	TITLE-ABS-KEY ((stroke OR “cerebrovascular accident” OR “brain infarction” OR “ischemic stroke” OR “hemorrhagic stroke” OR “brain attack”) AND (“early mobilization” OR “early mobilization” OR “early movement” OR “early rehabilitation”) AND (“clinical trial” OR “ECA” OR “randomized controlled trial” OR “randomized controlled trial” OR RCT))	182
Web of Science	TS = ((stroke OR “cerebrovascular accident” OR “brain infarction” OR “ischemic stroke” OR “hemorrhagic stroke” OR “brain attack”) AND (“early mobilization” OR “early mobilization” OR “early movement” OR “early rehabilitation”) AND (“clinical trial” OR “ECA” OR “randomized controlled trial” OR “randomized controlled trial” OR RCT))	

**Table 3 healthcare-14-00078-t003:** Characteristics and Results of the Selected Studies.

Author & Year	Sample Characteristics (Age, Sex, Stroke Type, Severity-NIHSS)	Population (n)	Experimental Group (Intervention)	Control Group (Comparison)	Outcomes & Assessments	Main Results
AVERT Group (2015) [47]	Mean Age: 72 yrs Sex: M 61%, F 39% Type: Ischemic 87.5%, Hemorrhagic 12.5%	n = 2104	Very Early Mobilization (VEM) • Started at ~18.5 h post-stroke. • Out-of-bed activities (sitting, standing, walking) + Standard Care (SC).	Standard Care (SC) • Started at ~23 h post-stroke. • Less frequent mobilization.	• Dependency (mRS) • Time to walk 50 m unassisted • Walking unassisted at 3 months • Adverse Events (AEs) & Mortality • ADL Independence (Barthel) • Assessments: Baseline (V0), 3 months (V1)	• mRS: Significant change favoring the Control Group at 3 months. • Walking ability: No significant differences between groups. • AEs & Mortality: No significant differences. • Barthel: Significant improvement in both groups, significantly greater in the Experimental Group (*p* < 0.001).
Cumming et al. (2019) [24]	Mean Age: 70 yrs Sex: M 58%, F 39% Type: Ischemic 87%, Hemorrhagic 13% Severity: Mild 55.5%, Moderate 30.5%, Severe 14%	n = 2104 EG: 1054 CG: 1050 (3-mo follow-up: n = 2031)	Very Early Mobilization (VEM) • Started at ~18 h post-stroke. • Duration: 31 min/day, 3 sessions/day. • Period: 14 days or until discharge.	Standard Care (SC) • Started at ~22 h post-stroke. • Duration: 10 min/day, 1 session/day. • Period: 14 days or until discharge.	• Quality of Life (AQoL-4D) • ADL Independence (Barthel) • Cognition (MoCA) • Depression/Anxiety (IDA) • Assessments: Baseline (V0), 3 months (V1), 12 months (V2)	• AQoL-4D (V0–V2): No significant differences between groups. • Correlation: Higher AQoL-4D was correlated with lower depression/anxiety, shorter length of stay, and higher MoCA and Barthel scores (*p* < 0.001).
Langhorne et al. (2017) [46]	Mean Age: 74 yrs Sex: M 58%, F 42% Type: Ischemic 87%, Hemorrhagic 13% Severity: Mild 53%, Moderate 28.5%, Severe 18.5%	n = 2104 EG: 1054 CG: 1052 (12-mo follow-up: n = 1797)	Very Early Mobilization (VEM) • Started at ~18 h post-stroke. • Duration: 31 min/day. • Period: 14 days or until discharge.	Standard Care (SC) • Started at ~23 h post-stroke. • Duration: 10 min/day. • Period: 14 days or until discharge.	• Dependency (mRS) • Days to walk 50 m unassisted • Adverse Events (AEs) • Cognition & Mood (IDA) • Quality of Life (AQoL) • Assessments: Discharge (V0), 3 months (V1), 12 months (V2)	• Dose–Response Analysis: Reported in the main AVERT trial. Outcomes consistent with AVERT 2015.
Yelnik et al. (2017) [50]	Mean Age: 66 yrs Sex: M 62%, F 38% Type: Ischemic 77.7%, Hemorrhagic 22.3% Severity: Mild 9.2%, Moderate 20.4%, Severe 20% % does not total 100	n = 103 EG: 52 CG: 51	High-Intensity Exercises (resistive movements, repetitions to fatigue, 45 min) + Low-Intensity Exercises (LIE) (15–20 min). • Started at ~55 h post-stroke. • Period: 10 sessions.	Low-Intensity Exercises (LIE) (passive movements, sitting, walking) 15–20 min. • Started at ~53 h post-stroke. • Period: 10 sessions.	• Motor Recovery (FMA) • Days to walk 10 m unassisted • Balance (PASS) • Dependency (mRS) • Function (FIM) • Quality of Life (SIS) • Adverse Events (AEs) • Assessments: V0 (Baseline), V2, V4	• FMA & PASS (V0–V4): Significant improvement in both groups (*p* < 0.001), no significant differences between groups. • Other Outcomes (V0–V4): No significant between-group differences for gait initiation, mRS, FIM, or SIS. • AEs: Frequent, but no significant differences between groups (EG: more falls; CG: more epilepsy).
Anjos et al. (2022) [52]	Mean Age: 60 yrs Sex: Not Specified Type: Post-thrombolysis Severity: Not Specified	n = 104 EG: 51 CG: 53	Very Early Mobilization (VEM) • Started <12 h post-thrombolysis. • Included glute bridge, sitting on edge of bed, standing, gait, and functional activities for upper limbs.	Standard Care (SC) • Started <24 h post-thrombolysis. • Included active bed exercises, postural correction, sitting balance, and gait training.	• Dependency (mRS) • Functional Mobility (TUG) • Balance (Berg) • Adverse Events (AEs) • Length of Stay, Mortality • Assessments: V0 (Baseline), V1, V2	• All Outcomes (V0–V2): No significant differences between groups for mRS, TUG, BERG, complications, length of stay, or mortality.
Rahayu et al. (2019) [51]	Mean Age: 57 yrs Sex: M 67.5%, F 32.5% Type & Severity: Not Specified	n = 40 EG: 20 CG: 20	Early Mobilization • Started at 24 h post-stroke. • Progressive protocol over 7 days (sensory stimulation, passive/active-assisted movements, postural control, sitting, standing, balance, functional tasks).	Later Mobilization • Started at 48 h post-stroke. • Slower progression of the same protocol.	• Balance (Berg Balance Scale) • ADL Independence (Barthel Index) • Assessments: V0 (Day 1), V1 (Day 5), V2 (Day 7)	• Berg (V0–V2): Significant difference favoring the Experimental Group (*p* = 0.038). • Barthel (V0–V2): Significant difference favoring the Experimental Group (*p* = 0.002 at V1, *p* = 0.021 at V2).
Tong et al. (2019) [48]	Mean Age: 61 yrs Sex: Not Specified Type: Ischemic 100% Severity: Mild 68.8%, Moderate 31.2% All patients post-thrombolysis	n = 248 G1: 80 G2: 82 G3: 86	**G2:** Thrombolysis + Early Mobilization (24–48 h). • Duration: <1.5 h/day. • Period: 10–14 days.	**G1:** Thrombolysis + Later Mobilization (>48 h). **G3:** Thrombolysis + Very Early Mobilization (<24 h). • Duration: ≥3 h/day. • Period: 10–14 days.	• Dependency (mRS) • Mean Training Time (min/day) • Time to start treatment (hours) • Assessments: Discharge (V0), 3 months (V1)	• mRS (V0–V1): Significant differences favoring G2 over G1 and G3 (*p* = 0.041). • Training Time: Significantly longer in G3 (184.6 min) and G2 (184.1 min) vs. G1 (53.4 min) (*p* < 0.001). **• Time to Start:** Significantly shorter in G3 (16.8 h) vs. G1 (41.0 h) and G2 (38.0 h) (*p* < 0.001).
Wang et al. (2022) [49]	Mean Age: 61 yrs Sex: M 59%, F 41% Type: Ischemic 100% Severity: Mild 33.6%, Moderate 66.4%	n = 110 EG: 56 CG: 54	Early Mobilization + Medication • Started at 24–48 h post-stroke. • Included turning, sitting, stretching, standing, sit-to-stand, balance, coordination, motor skills. • Duration: 20 min/session, 2 for/day.	Standard Care + Medication • Started at 72–96 h post-stroke. • Duration: 20 min/session, 2 for/day.	• Dependency (mRS) • Motor Function (FMA-UL, FMA-LL) • Length of Hospitalization • Assessments: V0 (Baseline), V1 (7 days), V2 (30 days), V3 (90 days)	• mRS (V2): Statistically significant change favoring the Experimental Group (*p* = 0.005). • FMA-UL: No significant between-group differences. • FMA-LL (V0–V1): Difference favored EG but was not significant (*p* = 0.07). No differences at V2–V3. • Both groups improved continuously over 3 months (*p* < 0.0001).

Abbreviations: M: Male; F: Female; NIHSS: National Institutes of Health Stroke Scale; n: sample size; EG: Experimental Group; CG: Control Group; VEM: Very Early Mobilization; SC: Standard Care; h: hours; min: minutes; ADL: Activities of Daily Living; mRS: modified Rankin Scale; FMA: Fugl-Meyer Assessment; PASS: Postural Assessment Scale for Stroke; FIM: Functional Independence Measure; SIS: Stroke Impact Scale; AQoL-4D: Assessment of Quality of Life-4 Domains; MoCA: Montreal Cognitive Assessment; IDA: Index of Depression and Anxiety; TUG: Timed Up and Go Test; FMA-UL: Fugl-Meyer Assessment for Upper Limb; FMA-LL: Fugl-Meyer Assessment for Lower Limb; AEs: Adverse Events.

**Table 4 healthcare-14-00078-t004:** Methodological quality of the articles evaluated using the PEDro scale.

Author/Item	C1	C2	C3	C4	C5	C6	C7	C8	C9	C10	C11	Total
Yelnik et al. 2017 [50]	X	X		X				X		X	X	7/10
Cumming et al. 2019 [24]	X	X	X	X				X	X	X	X	8/10
Langhorne et al. 2017 [46]	X	X	X	X				X	X	X	X	8/10
Anjos et al. 2022 [52]	X	X	X					X		X	X	6/10
Rahayu et al. 2019 [51]	X	X		X				X	X	X	X	6/10
Tong et al. 2019 [48]	X	X	X	X	X			X		X	X	6/10
Wang et al. 2022 [49]	X	X	X	X				X		X	X	7/10
AVERT group, 2015 [47]	X	X	X	X	X	X	X	X	X	X	X	10/10
Chippala et al. 2016 [53]	X	X	X	X				X		X	X	7/10

C1: eligibility criteria; C2: random allocation; C3: concealment of allocation; C4: group similarity at baseline; C5: blinding of subjects; C6: blinding of therapists; C7: blinding of assessors; C8: one key outcome obtained from 85% of subjects; C9: intention to treat analysis; C10: between group statistical comparisons; C11: point and variability measures for at least 1 key outcome. Criteria 2–11 scored.

## Data Availability

The data presented in this study are available from the corresponding author upon reasonable request. Due to ethical and privacy restrictions, the data cannot be made publicly accessible. The review was registered with the Open Science Framework (OSF) with the registration code https://osf.io/kvug4 (accessed on 1 December 2025).

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
