# Peer review of "Efficacy of Early Mobilization in Stroke Patients in Relation to Quality of Life and Level of Dependency: A Systematic Review"

_healthcare, 2025, doi:10.3390/healthcare14010078_

Round 1
Reviewer 1 Report
Comments and Suggestions for Authors
This systematic review aimed to analyze the effectiveness of early mobilization (EM) on quality of life and dependency levels in stroke patients. Additionally, the impact on anxiety and depression, the occurrence of adverse effects, and length of hospital stay were assessed. Findings indicated that EM performed within 24–48 hours post-stroke reduces dependency levels; however, no significant improvement in quality of life was observed. Evidence regarding anxiety and depression was inconclusive, and no significant differences were reported between groups concerning adverse events or reduction in hospital stay duration. This review demonstrates that EM is beneficial for reducing dependency after stroke, but there is no evidence of a significant improvement in quality of life. Further research is needed to establish clear protocols and appropriate intervention doses.
- In Line 165-167, ‘The PEDro scale was used to assess methodological quality, scoring 10 items related to internal validity and statistical analysis. Studies were classified as low’ The description of the methodological quality assessment criteria is insufficient. The original text only states that studies with a PEDro score lower than 6 were excluded, but it does not explicitly explain the basis for selecting this cutoff point. Please supplement the text with the rationale for the threshold score as prescribed by the original scale.
- In Line 167-169, ‘Risk of bias was evaluated using Cochrane’s tool across six domains: randomization, allocation concealment, blinding, incomplete outcome data, selective reporting, and other biases. Each domain was rated as low, high, or unclear risk, with justification provided [45]’. Regarding the mention of "other biases," please specify and supplement exactly which situations or factors constitute "other biases" in this context.
- In Line 106, ‘A systematic review was conducted following the PRISMA guidelines (Preferred Reporting Items for Systematic Reviews and Meta-Analyses) to synthesize current evidence on early mobilization in stroke patients regarding quality of life and dependency levels.’ The PRISMA statement requires reporting the methods used for synthesizing results. The article does not specify whether a meta-analysis was planned. If a meta-analysis was not conducted, the reasons should be explained (e.g., excessive heterogeneity among studies). If a meta-analysis was conducted, detailed descriptions of the model (fixed-effects vs. random-effects), effect measures (e.g., Risk Ratio [RR], Mean Difference [MD]), and heterogeneity testing methods (such as the I2 statistic) are required.
- In Line 329-330, ‘The findings across studies were heterogeneous and dependent on the 329 timing of reassessment and the timing of EM initiation following stroke’ The article mentions that "the findings across studies were heterogeneous," but it fails to provide a quantitative assessment of this heterogeneity.
- In Line 332-335, ‘Rahayu et al. (2019) reported a positive correlation within 7 days between EM 331 and a lower level of dependence. At 3 months post-treatment, three studies found no significant differences between groups’ When comparing results from different studies, the text only describes them as "significant" or "not significant" without considering the impact of sample size on statistical power. For instance, the study by Rahayu et al. had a small sample size (n=40), which may indicate insufficient statistical power.
- In Line 336-337, ‘The only study that conducted a 12-month follow-up for this variable was Langhorne et al. (2017) [46], who reported a significant difference between the experimental and control groups in favor of early mobilization in post-stroke patients.’ The article mentions that Langhorne et al. found a significant difference at the 12-month follow-up, but it does not report the specific effect size.
- In Line 148, Table 1. PICOS item generated keywords. Table 1: In the table, the PICOS items and the Keywords do not fully correspond. Specifically, the Keywords do not include content related to the Outcome.
Author Response
Author's Reply to the Review Report (Reviewer 1)
Comments 1:
- In Line 165-167, ‘The PEDro scale was used to assess methodological quality, scoring 10 items related to internal validity and statistical analysis. Studies were classified as low’ The description of the methodological quality assessment criteria is insufficient. The original text only states that studies with a PEDro score lower than 6 were excluded, but it does not explicitly explain the basis for selecting this cutoff point. Please supplement the text with the rationale for the threshold score as prescribed by the original scale.
Response 1: Dear Reviewer, thank you very much for emphasizing this section. This point has been revised to provide a clearer and more effective explanation, which improves its overall understanding.
“According to the PEDro scale studies were classified as low (<4), fair (4–5), good (6–8), or excellent (9–10). All studies with a methodological quality score of less than 6 were excluded from this systematic review.”
Comments 2:
- In Line 167-169, ‘Risk of bias was evaluated using Cochrane’s tool across six domains: randomization, allocation concealment, blinding, incomplete outcome data, selective reporting, and other biases. Each domain was rated as low, high, or unclear risk, with justification provided [45]’. Regarding the mention of "other biases," please specify and supplement exactly which situations or factors constitute "other biases" in this context.
Response 2: Thank you very much for this observation. This item is part of a standard domain within the Cochrane tool. However, the ‘other biases’ section has been removed to avoid potential misunderstandings.
Comments 3:
- In Line 106, ‘A systematic review was conducted following the PRISMA guidelines (Preferred Reporting Items for Systematic Reviews and Meta-Analyses) to synthesize current evidence on early mobilization in stroke patients regarding quality of life and dependency levels.’ The PRISMA statement requires reporting the methods used for synthesizing results. The article does not specify whether a meta-analysis was planned. If a meta-analysis was not conducted, the reasons should be explained (e.g., excessive heterogeneity among studies). If a meta-analysis was conducted, detailed descriptions of the model (fixed-effects vs. random-effects), effect measures (e.g., Risk Ratio [RR], Mean Difference [MD]), and heterogeneity testing methods (such as the I2 statistic) are required.
Response 3: Dear Reviewer, thank you very much for pointing out this issue. The text has been revised to avoid any confusion regarding the meta-analysis section, as follows: “This systematic review was conducted in accordance with Cochrane Collaboration recommendations and reported following the PRISMA 2020 statement to synthesize current evidence on early mobilization in stroke patients regarding quality of life and dependency levels”
Comments 4:
- In Line 329-330, ‘The findings across studies were heterogeneous and dependent on the 329 timing of reassessment and the timing of EM initiation following stroke’ The article mentions that "the findings across studies were heterogeneous," but it fails to provide a quantitative assessment of this heterogeneity.
Response 4: Dear Reviewer, thank you very much for this remark. You are correct that we are not conducting a meta-analysis. The text has been clarified to address this misunderstanding; we simply report differing results across the studies, as shown in the table.
Comments 5:
- In Line 332-335, ‘Rahayu et al. (2019) reported a positive correlation within 7 days between EM 331 and a lower level of dependence. At 3 months post-treatment, three studies found no significant differences between groups’ When comparing results from different studies, the text only describes them as "significant" or "not significant" without considering the impact of sample size on statistical power. For instance, the study by Rahayu et al. had a small sample size (n=40), which may indicate insufficient statistical power.
Response 5: Dear Reviewer, this section has been revised to incorporate your comment. Thank you very much
Comments 6:
- In Line 336-337, ‘The only study that conducted a 12-month follow-up for this variable was Langhorne et al. (2017) [46], who reported a significant difference between the experimental and control groups in favor of early mobilization in post-stroke patients.’ The article mentions that Langhorne et al. found a significant difference at the 12-month follow-up, but it does not report the specific effect size.
Response 6: Dear Reviewer, this section has been revised to incorporate your comment. Thank you very much
Comments 7:
- In Line 148, Table 1. PICOS item generated keywords. Table 1: In the table, the PICOS items and the Keywords do not fully correspond. Specifically, the Keywords do not include content related to the Outcome.
Response 7: Thank you for this contribution. Two additional keywords relevant to the search have been included, as shown in Table 2.
cerebrovascular accident/brain infarction
Reviewer 2 Report
Comments and Suggestions for Authors
Thank you for submitting your systematic review titled “Effectiveness of Early Mobilization in Stroke Patients in Relation to Quality of Life and Level of Dependency.” This is a well-conducted and timely review that addresses an important clinical question using rigorous methodology. The manuscript is clearly structured, follows PRISMA guidelines, and provides a balanced synthesis of current evidence. The conclusions are appropriately cautious and relevant for clinical practice and future research.
Specific Comments:
-
Clarity of Tables and Figures:
-
Table 3 (Study Characteristics and Results) requires careful formatting to ensure all columns align properly and the content is easily readable. In its current form, some headers and data appear misaligned, which may confuse readers.
-
Figures 4 and 5 (Risk of Bias per Article) reference domains “D1–D5” without explanation. Please add a brief legend or footnote defining each domain (e.g., D1: Randomization process, D2: Deviations from intended interventions, etc.) to aid interpretation.
-
-
Minor Revisions:
-
Please conduct a final proofread for minor formatting issues (e.g., repeated superscript “⁴” in author affiliations, stray index numbers in the text such as “240,” and the mention of Herisson et al. 2016 in the PEDro table despite its earlier exclusion).
-
In the Abstract, consider briefly stating the total number of studies/participants included for greater immediacy.
-
-
Discussion Strengths:
-
The discussion effectively contextualizes the findings, especially regarding the dissociation between functional improvement (dependency) and quality of life, and the possible physiological explanations for heterogeneous results (e.g., timing and dose of EM).
-
The section on healthcare costs and implications for the Spanish health system is a valuable addition that enhances the paper’s relevance.
-
Suggestions for Improvement:
-
Consider adding a small subsection in the Discussion or Limitations on the generalizability of findings, given that most included patients had ischemic stroke and mild-to-moderate severity.
-
While the narrative synthesis is clear, a summary table or figure visually depicting the direction of effect for each outcome (e.g., dependency, QoL, safety) across studies could further enhance readability.
The manuscript is generally well-written in clear and comprehensible English. The scientific narrative is logical, and terminology is used consistently and appropriately throughout.
Minor Issues for Correction:
-
Article Usage: There are occasional minor omissions or inconsistencies with articles ("a," "an," "the").
-
Example (Abstract): "this systematic review aimed to analyze the effectiveness of early mobilization (EM) on quality of life and dependency levels..."
-
Suggestion: "This systematic review aimed..." or "...the quality of life and the dependency levels..."
-
-
Capitalization: Standard title capitalization is not always followed in section headers.
-
Example: "3.7.2. Level of Dependence" should likely be "3.7.2. Level of Dependence" or formatted per journal style.
-
-
Minor Typos/Formatting Glitches: A careful final proofread is recommended to catch minor errors.
-
Example (Page 1, Author Affiliations): The superscript "⁴" appears repeated ("³⁴⁴⁴").
-
Example (Page 5, Search Strategy Table): "Web of OR “brain attack”" appears to be a formatting artifact.
-
Example (Page 6, Flow Diagram): The line "Documents excluded by title/abstract and/or study design (n =" is incomplete.
-
-
Sentence Flow (Minor): A few sentences are slightly long or could be streamlined for even greater clarity.
-
Example (Introduction, Page 2): "Rehabilitation is essential from early stages to chronic phases. Neuroplasticity underpins recovery, requiring individualized physiotherapy goals."
-
This is acceptable, but could be slightly smoothed: "Rehabilitation, essential from early to chronic phases, is underpinned by neuroplasticity and requires individualized physiotherapy goals."
-
Author Response
Comments 1:
- Clarity of Tables and Figures:
- Table 3 (Study Characteristics and Results)requires careful formatting to ensure all columns align properly and the content is easily readable. In its current form, some headers and data appear misaligned, which may confuse readers.
- Figures 4 and 5 (Risk of Bias per Article)reference domains “D1–D5” without explanation. Please add a brief legend or footnote defining each domain (e.g., D1: Randomization process, D2: Deviations from intended interventions, etc.) to aid interpretation.
Response 1: Dear Reviewer, thank you very much for your feedback. The tables have been revised to improve their clarity. Regarding the legends, they are clearly displayed on the right side of the table, identifying D1, D2, and so forth.
Comments 2:
Minor Revisions:
- Please conduct a final proofread for minor formatting issues (e.g., repeated superscript “⁴” in author affiliations, stray index numbers in the text such as “240,” and the mention of Herisson et al. 2016 in the PEDro table despite its earlier exclusion).
- In the Abstract, consider briefly stating the total number of studies/participants included for greater immediacy.
Response 2: Thank you for this note. All items have been corrected and the abstract has been improved. The article you mentioned has been removed from the table.
Comments 3:
- Discussion Strengths:
- The discussion effectively contextualizes the findings, especially regarding the dissociation between functional improvement (dependency) and quality of life, and the possible physiological explanations for heterogeneous results (e.g., timing and dose of EM).
- The section on healthcare costs and implications for the Spanish health system is a valuable addition that enhances the paper’s relevance.
Suggestions for Improvement:
- Consider adding a small subsection in the Discussion or Limitations on the generalizabilityof findings, given that most included patients had ischemic stroke and mild-to-moderate severity.
- While the narrative synthesis is clear, a summary table or figure visually depicting the direction of effect for each outcome (e.g., dependency, QoL, safety) across studies could further enhance readability.
Response 3: Thank you for your comment. The following paragraph has been added to the Limitations section: The generalizability of our results is limited by the patient cohort, which predominantly consisted of individuals with mild to moderate ischemic stroke. Therefore, our findings may not be directly applicable to patients with severe strokes or other stroke subtypes (e.g., hemorrhagic).reported following the PRISMA 2020 statement to synthesize current evidence on early mobilization in stroke patients regarding quality of life and dependency levels”
Comments 4:
- In Line 329-330, ‘The findings across studies were heterogeneous and dependent on the 329 timing of reassessment and the timing of EM initiation following stroke’ The article mentions that "the findings across studies were heterogeneous," but it fails to provide a quantitative assessment of this heterogeneity.
Response 4: Dear Reviewer, thank you very much for this remark. You are correct that we are not conducting a meta-analysis. The text has been clarified to address this misunderstanding; we simply report differing results across the studies, as shown in the table.
Comments 5:
Minor Issues for Correction:
- Article Usage:There are occasional minor omissions or inconsistencies with articles ("a," "an," "the").
- Example (Abstract): "this systematic review aimed to analyze the effectiveness of early mobilization (EM) on quality of life and dependency levels..."
- Suggestion: "Thissystematic review aimed..." or "...the quality of life and the dependency levels..."
Response: Corrected
- Capitalization:Standard title capitalization is not always followed in section headers.
- Example: "3.7.2. Level of Dependence"should likely be "3.7.2. Level of Dependence" or formatted per journal style.
- Minor Typos/Formatting Glitches:A careful final proofread is recommended to catch minor errors.
- Example (Page 1, Author Affiliations): The superscript "⁴" appears repeated ("³⁴⁴⁴").
- Example (Page 5, Search Strategy Table): "Web of OR “brain attack”"appears to be a formatting artifact.
- Example (Page 6, Flow Diagram): The line "Documents excluded by title/abstract and/or study design (n ="is incomplete.
Response: Corrected
- Sentence Flow (Minor):A few sentences are slightly long or could be streamlined for even greater clarity.
- Example (Introduction, Page 2): "Rehabilitation is essential from early stages to chronic phases. Neuroplasticity underpins recovery, requiring individualized physiotherapy goals."
- This is acceptable, but could be slightly smoothed: "Rehabilitation, essential from early to chronic phases, is underpinned by neuroplasticity and requires individualized physiotherapy goals."
Response: Corrected
Reviewer 3 Report
Comments and Suggestions for Authors
Dear authors. The article is devoted to an urgent topic - stroke, which occupies one of the dominant places in the structure of population mortality. The research approach is quite interesting, but together with tme it raises a number of questions.: The design of the study is not described, the criteria for inclusion/exclusion of articles are not clearly indicated, 3 publicly available databases are being analyzed, which is limited. And then it is necessary to provide arguments, or replace /expand the study. The novelty of the data obtained is also questionable: previously, quite a lot of work has been devoted to immobilization in stroke, while the authors draw the disappointing conclusion that this approach has no therapeutic advantages. In this regard (returning to the design of the study), it is advisable to consider immobilization only as a component of treatment, without putting it in the main position. Also, see the notes in the attached file.

Author Response
Comments 1:
Dear authors. The article is devoted to an urgent topic - stroke, which occupies one of the dominant places in the structure of population mortality. The research approach is quite interesting, but together with tme it raises a number of questions.: The design of the study is not described, the criteria for inclusion/exclusion of articles are not clearly indicated, 3 publicly available databases are being analyzed, which is limited. And then it is necessary to provide arguments, or replace /expand the study. The novelty of the data obtained is also questionable: previously, quite a lot of work has been devoted to immobilization in stroke, while the authors draw the disappointing conclusion that this approach has no therapeutic advantages. In this regard (returning to the design of the study), it is advisable to consider immobilization only as a component of treatment, without putting it in the main position. Also, see the notes in the attached file.
Response 1: The study design is appropriately addressed at the beginning of the Materials and Methods section; this is a systematic review. Regarding the inclusion and exclusion criteria, they are clearly defined: randomized controlled trials with a PEDro score greater than 6 are included in this review, in addition to other criteria such as the publication year range, as specified on line 151. The selected databases (PubMed/MEDLINE, Scopus, and Web of Science) are the three most prominent within the medical field. While additional databases could be considered, a review would not be considered complete or rigorous without these three. Concerning immobilization, the key point we aim to highlight in this article is the importance of early immobilization. Our objective is to emphasize this because, in many hospitals in Spain, treatment often does not begin in the early phase but is delayed until the subacute stage. This is the central issue we wish to bring to light.
In addition, the appropriate revisions have been incorporated into the manuscript, as indicated by the red text.
Round 2
Reviewer 3 Report
Comments and Suggestions for Authors
Dear authors. All necessary changes have been made. There are no methodological errors or inaccuracies. The article can be accepted for publication after the text has been edited by the editorial team.
Author Response
Dear ,Reviewer
Thank you very much